# Effect of Microbial Reinforcement on Polyphenols in the Acetic Acid Fermentation of Shanxi-Aged Vinegar

**Peng Du** [1] , **Yingqi Li** [1], **Chenrui Zhen** [1], **Jia Song** [1,*] , **Jiayi Hou** [1], **Jia Gou** [1], **Xinyue Li** [1], **Sankuan Xie** [2], **Jingli Zhou** [3], **Yufeng Yan** [3], **Yu Zheng** [1] **and Min Wang** [1,*]

1   State Key Laboratory of Food Nutrition and Safety, Tianjin Engineering Research Center of Microbial Metabolism and Fermentation Process Control, College of Biotechnology, Tianjin University of Science & Technology, Tianjin 300457, China; dp9298@163.com (P.D.); liyingqi9710@163.com (Y.L.); zhenchenrui@mail.tust.edu.cn (C.Z.); houjiayi0618@126.com (J.H.); gj20020424@126.com (J.G.); lixinyue020205@163.com (X.L.); yuzheng@tust.edu.cn (Y.Z.)

2   China National Research Institute of Food & Fermentation Industries Co., Ltd., Beijing 100015, China; xskuan@mail.tust.edu.cn

3   Shanxi Province Key Laboratory of Vinegar Fermentation Science and Engineering, Shanxi Zilin Vinegar Industry Co., Ltd., Taiyuan 030400, China; zhoujingli1110@126.com (J.Z.); yanyf8@163.com (Y.Y.)

*   Correspondence: tjsongjia@tust.edu.cn (J.S.); minw@tust.edu.cn (M.W.)

**Abstract:** Polyphenols are important functional substances produced in the acetic acid fermentation (AAF) of Shanxi aged vinegar (SAV). Previous studies have shown that the metabolic activity of microorganisms is closely related to polyphenol production and accumulation. In this study, microorganisms in the AAF of SAV were analyzed to explore how to increase the polyphenol yield by changing the microorganisms and reveal the potential mechanism of the microbial influence on the polyphenol yield. Macrotranscriptome analysis showed that acetic and lactic acid bacteria dominated the AAF fermentation process and initially increased and decreased. Spearman correlation analysis and verification experiments showed that the co-addition of *Acetobacter pasteurianus* and *Lactobacillus helveticus* promoted the accumulation of polyphenols, and the total polyphenol content increased by 72% after strengthening.

**Keywords:** Shanxi aged vinegar; polyphenols; acetic acid fermentation; microbial enhancement; *Acetobacter pasteurianus*

## 1. Introduction

Shanxi aged vinegar (SAV) has a long history of use as a condiment and a popular health food in China [1]. Sorghum, daqu, and bran are used for brewing SAV. SAV production includes the following stages. In semi-solid alcohol fermentation (AF), starch and proteins are hydrolyzed into glucose, ethanol, and other small-molecule compounds under the action of *Daqu* and environmental microorganisms. During the acetic acid fermentation (AAF), organic acids, such as acetic and lactic acid, are produced. In the Smoking *pei* (SP) stage (solid form), the mash after acetic acid fermentation is continuously heated and stirred daily. During the aging stage (in the liquid form), the vinegar is exposed to sunlight. Usually, AF, AAF, and SP take more than 20 days, while the aging stage may take decades or even centuries [2]. Due to its unique brewing process, SAV can produce many flavor substances and other functional compounds during fermentation. Polyphenols (PPs) are one of the main bioactive substances in plants and can be classified as flavonoids, non-flavonoids, and phenolic acids [3,4]. SAV is rich in PPs such as gallic acid, tannins, ferulic acid, catechin, quercetin, anthocyanins, p-coumaric acid, and resveratrol [3]. Polyphenols have a variety of biological activities and can prevent diseases such as cardiovascular diseases, liver injury, nerve degenerative diseases, cancer, and hyperlipidemia [5–9]. Various methods exist to obtain PPs from plant-based foods such as vegetables and grains [10,11]. They can also

be produced via chemical synthesis or biotransformation [12]. In addition, PPs can be biosynthesized de novo using microorganisms, such as fungi and bacteria, and microbial metabolites as precursors or catalysts [13].

Many types of microorganisms are involved in the AAF of SAV, including yeast, mold, acetic acid bacteria, lactic acid bacteria, and spore bacteria. During SAV fermentation, the species and abundances of microorganisms change constantly. Under the interaction and influence of these microorganisms, various compounds are metabolized and produced during fermentation [14]. The composition of microbial communities affects the composition and concentration of compounds in the AAF process and vice versa, as compounds also have essential effects on the growth and metabolism of microorganisms. Numerous studies have shown that various volatile and non-volatile compounds in SAV are produced by microbial activity [15,16]. By constructing high-yield microbial chassis cells, de novo microbial fermentation using a cheap carbon source as a substrate can effectively improve the efficiency of PPs synthesis [17]. Polyphenols are fermented using microbially modified synthases to yield high polyphenols [18]. However, our understanding of the distribution of PPs in the AAF process is limited, and research on the relationships between microorganisms and PPs in complex systems is scarce.

This study studied changes in the PPs in the AAF of the SAV phase using gas chromatography-mass spectrometry (GC-MS). Correlations between PPs, environmental factors, and microorganisms were analyzed and verified. There are few studies on SAV PPs, especially regarding the effect of microbial enhancement on key PPs in the AAF process. Therefore, this research aimed to study the main PPs and the effects of microbial enhancement on the AAF process using GC-MS to increase the accumulation of PPs in the SAV AAF process. The results of this study provide significant guidance for the production, operation, and quality optimization of SAV.

## 2. Materials and Methods

### 2.1. Microculture

#### 2.1.1. Strains

Regarding the strains used in the experiment, *Limosilactobacillus fermentum* SMR-360 (*Lac fer*) was identified as SMR-3601, SMR-3602, and SMR-3603, *Lactiplantibacillus plantarum* SMR-360 (*Lac pla*) was identified as SMR-3604 and SMR-3605, and *Lactobacillus helveticus* SMR-360 (*Lac hel*) was identified as SMR-3606, SMR-3607, SMR-3608, SMR-3609, SMR-3610, SMR-3611, SMR-3612, SMR-3613, SMR-3614, and SMR-3615. All strains were screened and stored at the Microbiology Laboratory of the School of Biological Engineering, Tianjin University of Science and Technology.

#### 2.1.2. Media

The lactic acid bacteria fermentation medium (MRS, *w/v*) consisted of glucose 2%, peptone 1%, beef extract 1%, yeast extract 0.5%, anhydrous sodium acetate 0.5%, Tween 80 0.1% (*v/v*), ammonium citrate 0.2%, dipotassium hydrogen phosphate 0.2%, magnesium sulfate 0.058%, and manganese sulfate 0.025%, with a pH of 6.2−6.8.

The acetic acid bacteria fermentation medium (GP, *w/v*) consisted of glucose 2%, peptone 2% and ethanol 5.0% (*v/v*). All chemical reagents used in this section were obtained from Beijing Boxbio Science & Technology Co., Ltd. (Beijing, China).

#### 2.1.3. Strain Culture

The strains were inoculated in the fermentation liquid medium. Lactobacilli were incubated in MRS medium at 37 °C for 24 h, and *Acetobacter* was incubated in GP medium at 30 °C with shaking at 180 rpm for 24 h. When the cell concentration reached $1 \times 10^8$ CFU/mL, it was used for subsequent experiments.

## 2.2. Sample Collection

Samples were collected from Shanxi Aged Vinegar Group Co., Ltd. In the AAF process, a uniform sample was obtained using methods described in the literature [19]. Vinegar-fermented grain samples were collected on the 1st, 3rd, 5th, 7th, and 9th day of AAF of SAV. Samples of acetamide were taken from a distance of approximately 30 cm from the surface of the sauce. All samples were collected using a five-point sampling method, with three copies of samples collected in parallel at each time point (i.e., one from three different tanks with the same brewing time), and each sample of the vinegar-fermented grains was approximately 200 g. Samples from the five-time points were mixed and placed in sterile Ziplock bags. The collected samples were immediately refrigerated in an icebox and stored in a refrigerator at −80 °C for subsequent use.

## 2.3. Analysis of PPs Contents and Composition during AAF

The PPs contents and composition analyses were carried out in the laboratory before microbial enhancement and at the Shanxi Aged Vinegar Group Co., Ltd. after microbial enhancement.

Before the assay, the solid vinegar paste sample was pretreated to obtain the extract. The solid sample (5 g) was added to 45 mL of water, shaken well for 3 h at room temperature, and centrifuged at 5000 rpm for 10 min (centrifuge TG16-WS, Xiangyi Co., Ltd., Changsha, China). Supernatants were collected for the analysis. The optimized forinophenol method was used to determine the PPs contents of the acetate samples.

The composition of PPs in the AAF phase samples was determined using GC-MS. First, 10 mL of the supernatant obtained in Section 2.2 was collected, and the PPs were extracted using the method described above [20]. Then, 2,4,5-trihydroxybenzoic acid was used as the internal standard, and 0.002 g was added to 10 mL of the extract during pretreatment. One milliliter of bis (trimethylsilyl) trifluoroacetamide (Supelco, Bellefonte, PA, USA) and 1% trimethylchlorosilane (Aladdin Biotechnology Co., Ltd., Shanghai, China) was added to the polyphenol extract, which was then reacted in a water bath at 70 °C for 3 h (DK-8D, Qiaofeng Co., Ltd., Shanghai, China). Compounds were identified by comparison with the standard's retention times and mass spectrometry results. Their concentrations were calculated by comparing the peak areas of the internal standard compounds.

## 2.4. Macrotranscriptome Sequencing of Microbial Communities during AAF

### 2.4.1. Extraction of RNA

Total RNA was extracted using the Total RNA Extraction Kit (Mobio, document number: 12866-25). Ribosomal RNA was removed using the Ribo-Zero Magnetic Gold Kit (Epidemiology, article number: MRZE724). The TruSeq Chain mRNA LT Sample Preparation Kit (Illumina, San Diego, CA, USA) was used for reverse transcription and library construction.

### 2.4.2. Library Quality Checks and Sorting

One microliter of the library was accurately obtained, and 2100 quality checks were performed on an Agilent Bioanalyzer using the Agilent Highly Sensitive Kit (Agilent Technologies Inc., Santa Clara, CA, USA). A qualified library should have a single peak and no adapters. Libraries were quantified on a Promega QuantiFluor using the Quant-it PicoGreen double-stranded DNA Analysis Kit. For qualified libraries, we performed $2 \times 150$ bp double-ended sequencing on a NextSeq machine (Personal Bio, Shanghai, China) using the NextSeq 500 High-Yield Kit (300 cycles). Quality control of the raw data was performed by sequencing using FASTQC (http://www.bioinformatics.babraham.ac.uk/projects/fastqc/, accessed on 23 June 2023). CutAdapt (v1.2.1) [21] was used to screen and filter raw data from the sequencing machines. SortMeRNA (http://ioinfo.lifl.fr/rna/sortmerna/, accessed on 23 June 2023) [22,23] was used to eliminate rRNA, and Trinity (http://trinityrnaseq.github.io/, accessed on 23 June 2023) [24] was used for sequence assembly and splicing. The redundancies with a similarity of 0.95 and minimum

coverage of 0.9 were merged and removed using a high-consistency tolerance cluster database, and the longest sequence was used as the representative sequence of UniGene to construct the UniGene set. The databases included NR, GO, KEGG, eggnog, CAXY and Swiss-Prot for Unigene feature annotation [23]. The metatranscriptome sequencing data used in this study were uploaded to the NCBI database under the registration numbers SRX7581352-SRX7581356.

### 2.5. Effects of Microorganisms on PPs

In the single-factor experiment carried out in the laboratory, the exogenous addition of microorganisms was used to control the microbial composition of vinegar-fermented grains. Exogenous microorganisms were added to a 500 mL conical flask containing 300 g of vinegar-fermented grains. The vinegar-fermented grains in the control group (Control) did not contain exogenous microorganisms. *Lactobacillus helveticus*, *Lactiplantibacillus plantarum*, and *Acetobacter pasteurianus* were added to the vinegar-fermented grains in the experimental groups. The addition amount and time were the same as those in the control group, and superior strains were selected. The additional amount of the superior strains was set to $10^8$ CFU/100 g of vinegar-fermented grains. On this basis, a suitable fermentation time was selected to evaluate the additional effect of microorganisms.

### 2.6. Determination of Physical and Chemical Indices during AAF

The pretreated samples were used to determine the physical and chemical indices. The pH was measured using a pH meter S20P (Mettler Toledo Company, Shanghai, China). The total acid content was determined by titration with a standard solution (0.1 mol/L sodium hydroxide) with phenol red as an indicator. The amino nitrogen content was determined using the ninhydrin method of the European winemaking convention. Changes in the pH, total acid content, and amino nitrogen content during AAF were detected using the following methods [25]. The total reduced sugar content was determined using a film test. A fully automatic Sykam amino acid analyzer S433D (Sykam Co., Ltd., Eresing, Germany) was employed for the qualitative and quantitative detection of amino acids in accordance with the manufacturer's instructions. The temperature at the sampling point was measured using a thermometer. The reduced sugar content of the sample was determined using high-performance liquid chromatography (Agilent 1260, Agilent Technologies Inc., Santa Clara, CA, USA) [26].

### 2.7. Statistical Analysis

R software (version 3.6.3)was used to draw a bar chart of the composition of the dominant species in each sample (the species with the top 20 overall expressions) at each classification level.

Based on the species-level composition spectrum of each sample annotated in the database, R software was used to calculate the number of common groups, and the number of common and unique species in each sample was visually presented using a Venn diagram (https://en.wikipedia.org/wiki/Venn_diagram, accessed on 23 June 2023).

The data for each sample were measured at least three times, and the experimental data were expressed as the mean ± standard deviation (mean ± SD). Spearman's correlations were analyzed using SPSS software (19.0). The correlations between microorganisms, environmental factors, and PPs were analyzed, and a heat map was drawn using Multiple Experiment Viewer software (version 4.9.0).

## 3. Results

*3.1. Correlation Analysis between PPs and Microorganisms during AAF of SAV*

3.1.1. Change in the PPs Contents during AAF

The total ion strength chromatogram is shown in Figure 1B,C. From the 1st to 9th day of AAF, the number of substances detected was $509 \pm 77$, $580 \pm 104$, $541 \pm 53$, $546 \pm 49$, and $483 \pm 11$, respectively (Figure 1B,C). The number of PPs species gradually increased from seven on the 1st day to nine on the 7th day and then decreased to seven on the 9th day (Figure 1A,D). The identity of the compounds was determined (Table 1). By searching the NIST20 database, 10 types of PPs were found in the AAF of SAV: (1) p-coumaric acid, (2) 4-hydroxy-3-methoxyphenylethylene Glycol, (3) vanillic acid, (4) ferulic acid, (5) L-epicatechin, (6) gallic acid, (7) 3-(3-hydroxy-4-methoxyphenyl)propanoic acid, (8) caffeic acid, (9) (E)-3-(3-hydroxyphenyl)acrylic acid ethyl ester, and (10) vanillin. The type and contents of PPs in the AAF samples differed with time. The PPs types gradually changed from the 1st to the 9th day. Polyphenols 1–4 were detected throughout the AAF process, while PP6 and PP10 were observed in the middle and later AAF stages, respectively. Polyphenol 8 and PP9 appeared on days 1 and 5 and 3 and 7 of the AAF process, respectively (Table 1). The PPs contents changed during AAF. The PP6 and PP10 contents increased gradually with increasing AAF time, whereas the PP1 contents first decreased and then increased. The PP2, 3, 4, 5, 7, 8, and 9 contents showed no apparent trends; however, the PP2, 3, and 5 contents at the end of AAF were higher than at the beginning.

During the AAF process, the total PPs contents gradually increased, reaching a maximum of $47.04 \pm 2.74$ mg/g of vinegar-fermented grains on the 9th day. Compared with the first day of AAF, the total PPs contents increased by 37.42% (Figure 1A).

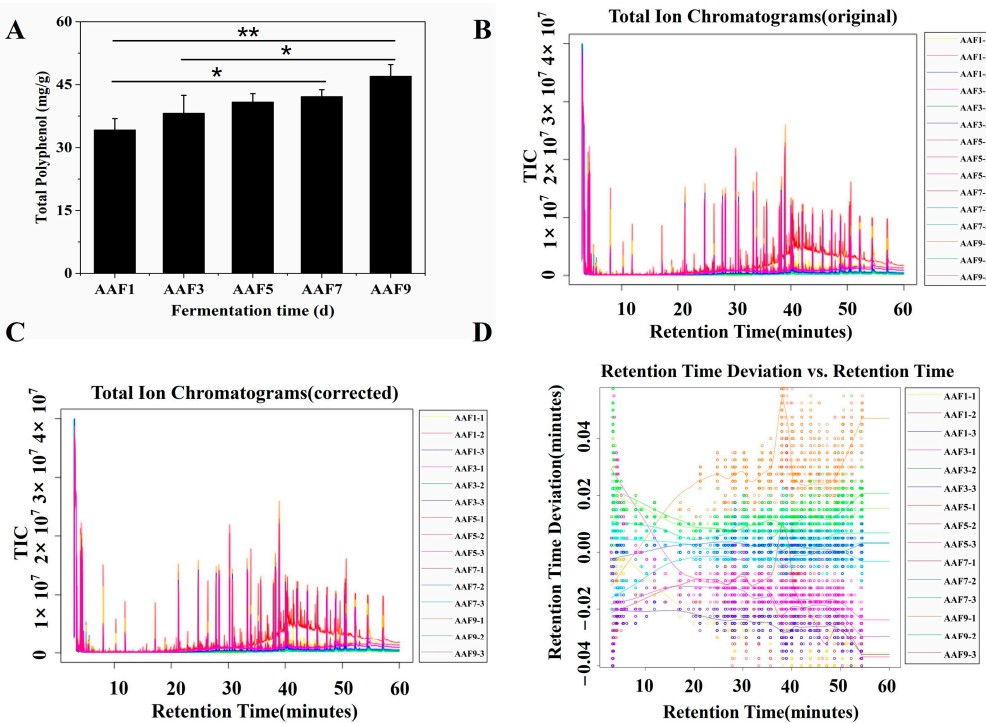

**Figure 1.** Analysis of polyphenol changes in the acetic acid fermentation (AAF) of Shanxi aged vinegar (SAV). (**A**): Change in the polyphenol contents during AAF; (**B–D**): gas chromatography–mass spectroscopy total ion diagram during the AAF of SAV; * $p < 0.05$; ** $p < 0.01$. TIC, total ion chromatograms.

**Table 1.** Distribution of polyphenols in the acetic acid fermentation (AAF) of Shanxi aged vinegar (SAV).

| Compound Identification | CAS No. | Relative Content (%) | | | | |
|---|---|---|---|---|---|---|
| | | AAF 1d | AAF 3d | AAF 5d | AAF 7d | AAF 9d |
| p-Coumaric acid | 501-98-4 | 0.102 ± 0.068 [ab] | 0.053 ± 0.037 [a] | 0.07 ± 0.06 [ab] | 0.127 ± 0.053 [ab] | 0.193 ± 0.017 [b] |
| 4-Hydroxy-3-methoxyphenylethylene Glycol | 534-82-7 | 0.156 ± 0.156 [a] | 0.074 ± 0.106 [a] | 0.013 ± 0.008 [a] | 0.067 ± 0.153 [a] | 0.0425 ± 0.047 [a] |
| Vanillic acid | 121-34-6 | 0.127 ± 0.213 [a] | 0.013 ± 0.017 [a] | 0.05 ± 0.02 [a] | 0.037 ± 0.017 [a] | 0.047 ± 0.028 [a] |
| Ferulic Acid | 1135-24-6 | 0.157 ± 0.077 [a] | 0.077 ± 0.073 [a] | 0.3 ± 0.08 [a] | 0.263 ± 0.147 [a] | 0.307 ± 0.083 [a] |
| L-Epicatechin | 490-46-0 | 0.301 ± 0.419 [a] | 0 | 0.075 ± 0.015 [a] | 0.11 ± 0.05 [a] | 0.053 ± 0.017 [a] |
| Gallic acid | 149-91-7 | 0 | 1.727 ± 0.3347 [a] | 2.59 ± 0.69 [a] | 2.383 ± 0.483 [a] | 2.865 ± 2.775 [a] |
| 3-(3-Hydroxy-4-methoxyphenyl)propanoic acid | 1135-15-5 | 0.163 ± 0.137 [a] | 0.053 ± 0.037 [a] | 0 | 0.107 ± 0.067 [a] | 0 |
| Caffeic acid | 331-39-5 | 0.063 ± 0.097 [a] | 0 | 0.05 ± 0.05 [a] | 0 | 0 |
| 2-Propenoic acid, 3-(3-hydroxyphenyl)-, ethyl ester, (2E)- | 96251-92-2 | 0 | 0.018 ± 0.022 [a] | 0 | 0.14 ± 0.11 [b] | 0 |
| Vanillin | 121-33-5 | 0 | 0 | 0 | 0.007 ± 0.007 [a] | 0.0175 ± 0.017 [a] |

Data are expressed as means ± standard deviations; different letters indicate significant differences among the groups ($p < 0.05$).

### 3.1.2. Changes in the Physical and Chemical Indices during AAF

Temperature, pH, acidity, reducing sugar, amino nitrogen, and alcohol contents were the key physical and chemical indices monitored during AAF. Figure 2 summarizes the analysis of the changes in the physical and chemical characteristics of AAF after the intensive experiments. The temperature increased at first, then decreased slightly, reaching a maximum of 44.73 °C on the third day (Figure 2A). The amino nitrogen and total acid contents increased gradually and reached a maximum on the 9th day, reaching the maximum values of 0.06 ± 0.012 and 3.94 ± 0.17 g/100 g of vinegar-fermented grains, respectively (Figure 2C,D). The reduced sugar content first increased and then decreased and reached the maximum on the 7th day, which was 2.45 ± 0.09 g/100 g of vinegar-fermented grains (Figure 2B). The alcohol content gradually decreased during the AAF process (Figure 2E). The pH fluctuated slightly in the range of 3.7–4.0, but there was no apparent trend (Figure 2F).

Sixteen amino acids were detected in the AAF culture samples: glutamic acid, glycine, alanine, valine, methionine, isoleucine, leucine, phenylalanine, threonine, aspartic acid, serine, proline, tyrosine, histidine, lysine, and arginine. Regarding the total amino acid concentration, there were significant differences among the different AAF stages. Changes in the contents of various amino acids during AAF are shown in the Supplementary Table S1, and with the exception of threonine, proline, histidine, and lysine, most increased after AAF. Glutamic acid, arginine, and aspartic acid levels significantly increased. From the 1st to the 9th day of AAF, the alanine content decreased slightly and then increased gradually. The tyrosine content increased rapidly from the 1st to the 3rd day, and then its increase slowed.

Six reducing sugars (Supplementary Table S2) were detected during AAF: xylose, fructose, mannose, glucose, maltose, and trehalose. The xylose and mannose contents increased gradually, while the fructose content first increased and then decreased before increasing again, while the trehalose content first increased and then decreased. Changes in the glucose and maltose contents were not obvious.

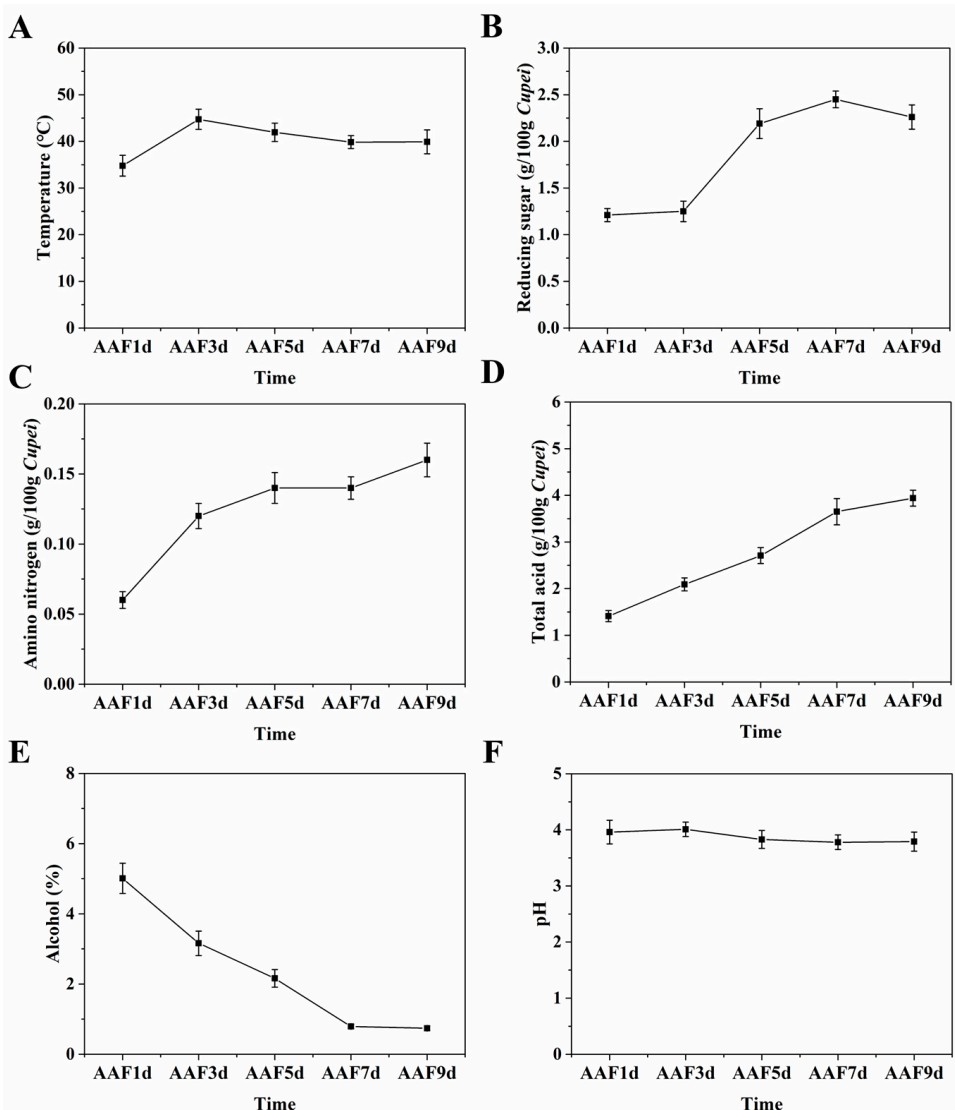

**Figure 2.** Changes in physicochemical indexes during the acetic acid fermentation (AAF) of Shanxi aged vinegar (SAV). (**A**–**F**): Changes in the temperature, reducing sugar, amino nitrogen, total acid, alcohol, and pH, respectively, over time. Values are means ± standard deviations.

### 3.1.3. Succession Law of the Microbial Community during AAF

During the AAF process, microorganisms play an essential role in accumulating PPs. Changes in microbial succession during AAF were analyzed (Figure 3). The number of microbial species constantly changed throughout the AAF process, with 133 species in total and 255, 22, 375, 28, and 62 species unique to AAF1d, 3d, 5d, 7d and 9d, respectively. Differences in the microbial species at different fermentation times were caused by the succession of open-fermentation microorganisms. The sum of the abundance of the top 20 most abundant species accounted for more than 98% of the total microbial abundance. Among these species, the abundance ratios of *Acetobacter bustiensis* in AAF1d-9d samples were: 0.49% (AAF1d), 0.04% (AAF3d), 12.23% (AAF5d), 13.02% (AAF7d), and 18.74% (AAF9d). The abundance ratios of *Lactobacillus* acid fast in AAF1d-9d samples were: 30.74% (AAF1d), 94.55% (AAF3d), 36.22% (AAF5d), 26.42% (AAF7d), and 16.49% (AAF9d), showing an initial increase followed by a decrease. *Lactobacillus acetotolerans* and *Acetobacter pasteurianus* were the dominant flora in the AAF process. Microorganisms with higher abundances included *Lactobacillus helveticus*, Unclassified *Lactobacillales*, and Unclassified *Lactobacillus*.

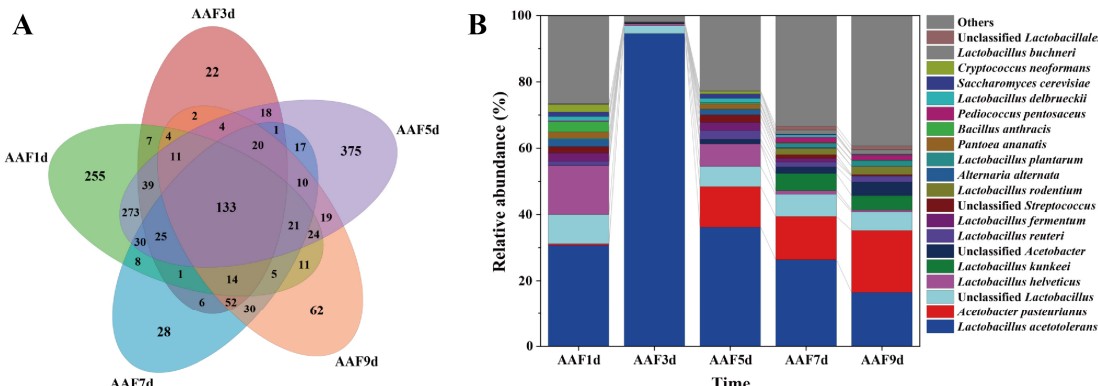

**Figure 3.** Microbial succession during the acetic acid fermentation (AAF) of Shanxi aged vinegar (SAV). (**A**): Microbial diversity during AAF in SAV; (**B**): changes in microbial abundance during AAF in SAV.

### 3.1.4. Correlation Analysis between PPs and Microorganisms during AAF

Microorganisms are essential factors affecting PPs in AAF. Using microorganisms as key environmental factors, Spearman's correlation analysis was performed, and correlations between microorganisms and PPs were analyzed (Figure 4). There were significant correlations between many types of microorganisms and PPs. Among these, *Lactobacillus helveticus* and *Cryptococcus neoformans* had significant positive correlations with PP5, PP3, and PP8, *Bacillus anthracis* had a significant positive correlation with PP5 and PP3, and a negative correlation with PP6, acetic acid bacteria had a significant positive correlation with PP4, and *Lactiplantibacillus plantarum*, *Lactobacillus rodentium*, and *Lentilactobacillus buchneri* had a significant positive correlation with PP1 and PP10. Therefore, *Lactobacilli* and acetate bacteria may be essential microorganisms affecting the formation of PPs by altering the production of PP1, 3, 4, 5, 8, and 10. We speculated that these may be the main microorganisms affecting PPs (Figure 4).

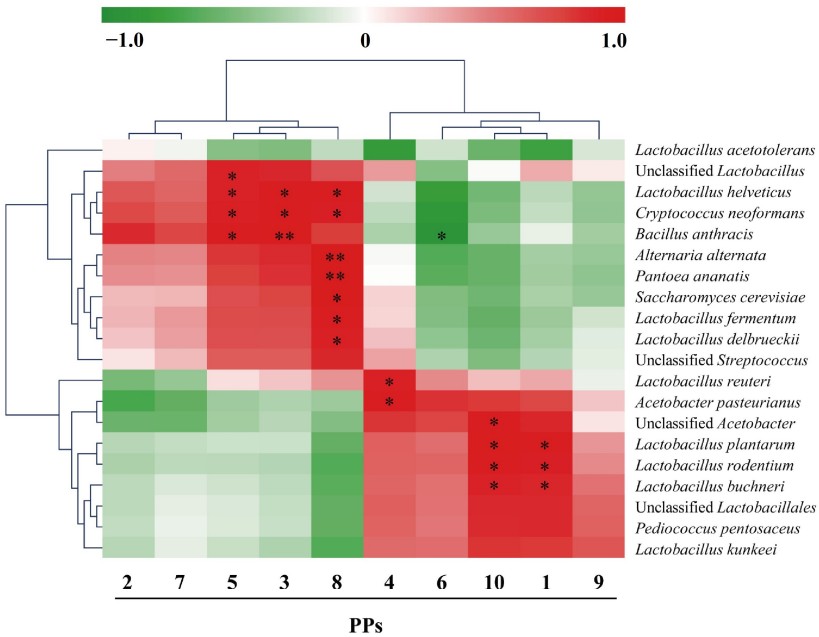

**Figure 4.** Spearman correlation analysis heat map between polyphenols (PPs) and microorganisms during the acetic acid fermentation (AAF) of Shanxi aged vinegar (SAV). Red and green indicate positive and negative correlations, respectively, and the shade suggests the size of the correlation coefficient; * $p < 0.05$; ** $p < 0.01$. Please refer to Section 3.1.1. for the PPs associated with each PP number.

### 3.2. Single Factor Verification of Microorganisms Affecting PPs Contents

The above analysis showed that microorganisms may be an essential factor affecting the total PPs content. The effect of microorganisms on the total PPs content was verified by a microbial single-factor screening experiment (Figure 5).

First, the strains producing high levels of PPs were screened experimentally, showing that the total PPs content was higher when *adding Lactobacillus helveticus*. Compared with the control group, exogenous microorganism addition promoted the accumulation of total PPs. On this basis, the microbial addition was screened, and the results showed that with the increase in microbial addition, the total PPs content gradually increased. The total PPs content was highest when *Lactobacillus helveticus* was added at $10^9$ CFU/100 g of vinegar-fermented grains or when *Acetobacter pasteurianus* was added at $10^7$ CFU/100 g of vinegar-fermented grains. Finally, the addition time was screened, and the total PPs content reached a maximum on the 3rd day of the AAF process. These results showed that adding exogenous microorganisms and increasing the number of microorganisms can increase the total PPs content in the AAF process. In summary, on the 3rd day of AAF, a mixture of *Lactobacillus helveticus* ($10^9$ CFU/100 g of vinegar-fermented grains) and *Acetobacter pasteurianus* ($10^7$ CFU/100 g of vinegar-fermented grains) maximized the accumulation of total PPs.

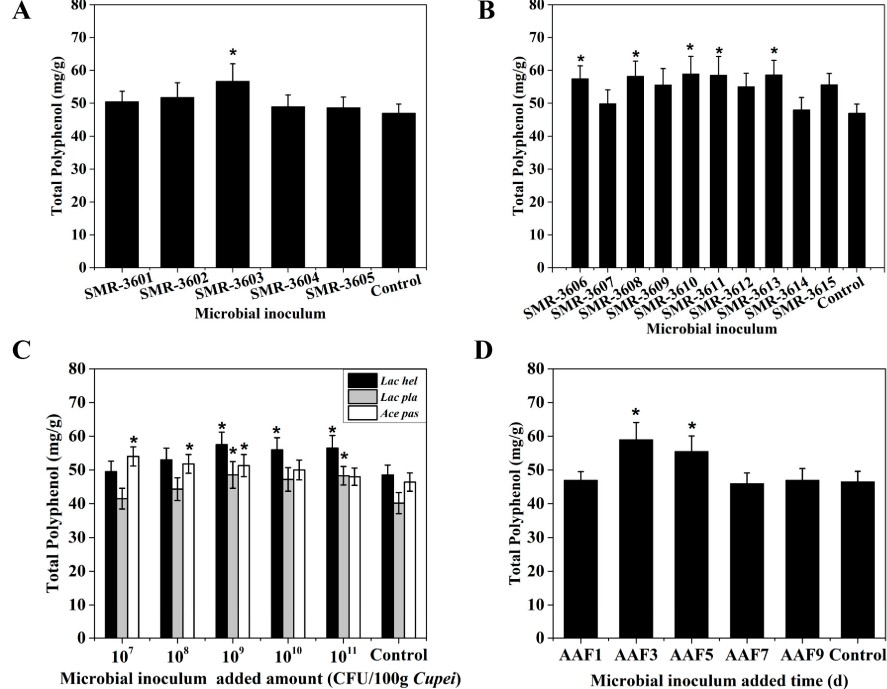

**Figure 5.** Screening microorganisms related to PPs during the acetic acid fermentation (AAF) of Shanxi aged vinegar (SAV). (**A,B**): Screening of microbial species; (**C**): screening of microbial addition amount; (**D**): screening of microbial addition time; * $p < 0.05$. *Lac hel*, *Lactobacillus helveticus*; *Lac pla*, *Lactiplantibacillus plantarum*; *Ace pas*, *Acetobacter pasteurianus*.

### 3.3. In Situ Regulation of PPs during AAF

Based on the effects of microbial factors on PPs accumulation, exogenous microbial enhancement experiments were performed. A schematic diagram of the GC × MS total ion current and a comparison of the PPs content before and after enhancement are shown in Table 2 and Figure 5. The results showed that the total number of detected substances was higher in the experimental group than in the control group.

After the intensive experiment, the PPs type and contents, reducing sugars and amino acid contents, and utilization rate of raw materials changed significantly during the AAF process (Figure 6, Supplementary Table S1). Twelve types of PPs were detected in the

experimental group, with (11) isoferulic acid and (12) salicylic acid detected in addition to the 10 PPs detected in the control group (Table 1). The distribution of PPs in the AAF process was analyzed after an intensive experiment. Like the control group, PP1, 4, and 6 were the main compounds in the experimental group. Compared with the control group, the contents of five PPs in the test group increased: PP1, 2, 4, 6, and 10. Among these, the largest increases were for PP4, 6, and 10. Therefore, the accumulation of PPs mainly increased through the promoted production of PP4, 6, 10, 11, and 12.

**Table 2.** Comparison of the polyphenol contents during the acetic acid fermentation (AAF) of Shanxi aged vinegar (SAV) before and after the enhancement experiment.

| | Compound Identification | CAS No. | Relative Content (%) | |
|---|---|---|---|---|
| | | | **Before Optimization** | **After Optimization** |
| 1 | p-Coumaric acid | 501-98-4 | $0.193 \pm 0.017$ | $0.256 \pm 0.019$ |
| 2 | 4-Hydroxy-3-methoxyphenylethylene Glycol | 534-82-7 | $0.0425 \pm 0.0047$ | $0.07 \pm 0.005$ |
| 3 | Vanillic acid | 121-34-6 | $0.047 \pm 0.028$ | $0.052 \pm 0.012$ |
| 4 | Ferulic Acid | 1135-24-6 | $0.307 \pm 0.083$ | $0.412 \pm 0.075$ * |
| 5 | L-Epicatechin | 490-46-0 | $0.053 \pm 0.017$ | $0.043 \pm 0.016$ |
| 6 | Gallic acid | 149-91-7 | $2.865 \pm 0.175$ | $4.1 \pm 0.21$ ** |
| 7 | 3-(3-Hydroxy-4-methoxyphenyl)propanoic acid | 1135-15-5 | 0 | 0 |
| 8 | Caffeic acid | 331-39-5 | 0 | 0 |
| 9 | 2-Propenoic acid, 3-(3-hydroxyphenyl)-, ethyl ester, (2E)- | 96251-92-2 | 0 | 0 |
| 10 | Vanillin | 121-33-5 | $0.0175 \pm 0.007$ | $0.23 \pm 0.009$ *** |
| 11 | Isoferulic acid | 1135-16-6 | 0 | $0.5 \pm 0.05$ *** |
| 12 | Salicylic acid | 69-72-7 | 0 | $0.4 \pm 0.05$ *** |

The data are expressed as means $\pm$ standard deviations, and the different symbols indicate significant differences between groups (* $p < 0.05$, ** $p < 0.01$, *** $p < 0.001$).

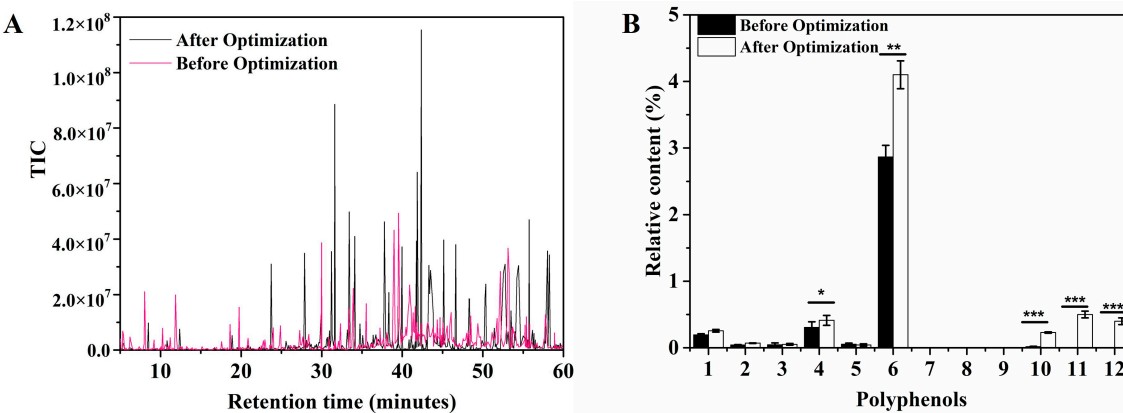

**Figure 6.** Gas chromatography–mass spectroscopy total ion diagram and polyphenols content during the acetic acid fermentation (AAF) of Shanxi aged vinegar (SAV) before and after the enhancement experiment. (**A**): Gas chromatography–mass spectroscopy total ion diagram. (**B**): Polyphenol contents. * $p < 0.05$, ** $p < 0.01$, *** $p < 0.001$. TIC, total ion chromatogram. Please refer to Sections 3.1.1 and 3.3. for the polyphenols associated with each number.

## 4. Discussion

Solid-state AAF is the primary process involved in SAV production and brewing. The SAV samples were analyzed during AAF. SAV is rich in amino acids and functional components, such as PPs [19]. The SAV brewing process involves natural fermentation, which leads to the growth of various microorganisms. These microorganisms have an important influence on vinegar quality. Polyphenols accumulate during the AAF process and are affected by various environmental factors, such as microorganisms. In this study, environmental factors affecting PPs, including microorganisms, were analyzed, providing new research data for increasing the yield of PPs.

Increased microbial diversity or abundance can enrich enzyme lineages, which may facilitate the synthesis of functional substances, such as PP4 and PP10 [27,28]. Therefore, adding microorganisms may be an effective means of increasing the polyphenol content. As our results of in situ regulation showed, microorganisms can increase the total PPs content, with the PP4 and PP10 contents increasing by 95.54% and 100%, respectively. PP8 disappeared at the end of the AAF. Simultaneously, the Spearman correlation analysis also showed that PP8 was significantly related to *Lactobacillus helveticus*, *Cryptococcus neoformans*, and other microorganisms. Previous studies have shown that the addition of microorganisms can enhance the activity of caffeic acid 3-O-methyltransferase in catalytic reactions, increase the methylation of PP8, and promote the transformation of PP8 to PP4, resulting in the accumulation of PP4. In addition, microorganisms can produce decarboxylases to decarboxylate PP4 into PP10 [29], which may be an essential reason for the increased accumulation of PP10 (and total PPs).

Furthermore, we speculated that microorganisms may indirectly promote the synthesis of PPs in the AAF process by affecting environmental, physical, and chemical conditions, such as the contents of reducing sugars and amino acids. As shown in Figure 6, the PP4, PP6, and PP10 contents increased significantly. Previous studies have shown that amino acids, particularly aromatic amino acids, are essential precursors for the synthesis of PPs. Therefore, the addition of exogenous microorganisms can promote the accumulation of PPs precursors. For example, free amino acids can be produced by degradation by lactic acid bacteria [30,31], such as tyrosine, which acts as a precursor that promotes PP6 [32]. Recent studies have shown that microorganisms promote PPs production by participating in free amino acid biosynthesis during AAF.

Moreover, the addition of exogenous microorganisms indirectly affects synthetic reactions by promoting the transformation of reducing sugars to PPs. For example, mannose can be a precursor to form PP4 under microbial catalysis [33]. Additionally, glucose and phenylalanine can be precursors for PP10 biosynthesis [34,35].

Some studies have shown that temperature increases in the early stage of AAF contribute to cell wall destruction, which promotes the release of intracellular or bound PPs and increases the content of PPs such as PP4. Simultaneously, high temperatures promote the transformation of PP4 into its corresponding isomers, which may also cause PP11 formation. In addition, some amino acids are non-neutral, which may cause changes in environmental factors such as pH, affecting the release of PPs. The Spearman correlation between PPs and environmental, physical, and chemical factors also showed that amino acids and reducing sugars were significantly correlated with the yield of PPs (PP4, 6, and 10; Supplementary Figure S1).

Macrotranscriptomes were used to analyze the distribution of microorganisms and their effects on vinegar community succession during AAF. As shown in Figure 2, bacterial diversity increased at the initial fermentation stage and decreased gradually in the later stages, consistent with previous community studies. The Spearman correlation analysis between PPs and microorganisms also showed that acetic acid and lactic acid bacteria were significantly correlated with PPs accumulation. Although PP8 was significantly correlated with various microorganisms, it has little influence on vinegar's nutritional and health functions; therefore, it was not considered. In addition, *Lactobacillus rodentium*, *Lactobacillus buchneri*, and other strains were not detected in the fermented grains of SAV vinegar. Therefore, *Lactobacillus helveticus*, *Lactiplantibacillus plantarum* and *Acetobacter pasteurianus* were selected as the dominant strains for the strengthening experiments to improve the PPs yield. The results showed a strong correlation between the microorganisms and PPs during AAF. The metabolic activities of microorganisms play key roles during the AAF stage. Microbial diversity and flavor formation in SAV during fermentation is very different from those of other vinegars [36]. SAV has a unique microbial community structure, phenolic acid content, and physicochemical properties [37].

This study used Spearman correlation analysis and single-factor experiments to screen and verify microbial factors. Based on the effects of microorganisms on PPs, regulatory

strategies to promote the accumulation of PPs are proposed from the perspective of microbial enhancement. First, the distribution of the PPs was analyzed. Ten PPs were detected during the AAF of SAV. The effect of microorganisms on PPs yield was observed. By adding exogenous microorganisms, it was verified that a mixture of lactic and acetic acid bacteria could increase the accumulation of PPs. Although microbial factors promote the accumulation of PPs, the mechanism of action of PPs in AAF requires further study. In the future, we will simulate AAF conditions, build a reaction model, and comprehensively study the formation mechanism of PPs in the AAF process.

## 5. Conclusions

Microorganisms are essential factors that affect the accumulation of PPs during AAF. Metatranscriptomic analysis revealed that acetic and lactic acid bacteria were the dominant bacteria affecting the PPs yield and participating in regulating the AAF process. During fermentation, acetic acid bacteria gradually increase, whereas lactic acid bacteria increase and decrease. Validation experiments with the exogenous addition of *Acetobacter pasteurianus* and *Lactobacillus helveticus* also showed that the total PPs content increased by 72% after their addition. Therefore, in the SAV AAF process, the production of PPs can be increased by the exogenous addition of a mixture of *Acetobacter pasteurianus* and *Lactobacillus helveticus*.

**Supplementary Materials:** The following supporting information can be downloaded at: https://www.mdpi.com/article/10.3390/fermentation9080756/s1, Figure S1: Spearman correlation between physicochemical and polyphenols. Red and green indicate positive correlation and negative correlation, respectively, and the shadow of color indicates the correlation coefficient between environmental factors and polyphenols; Table S1: Changes of amino acid contents in the AAF of SAV; Table S2: Changes of reducing sugar content in the AAF of SAV.

**Author Contributions:** Conceptualization, P.D. and Y.L.; methodology, P.D. and C.Z.; software, J.S.; validation, J.H., Y.Y. and J.G.; formal analysis, X.L.; investigation, S.X.; resources, J.Z.; data curation, Y.Z.; writing—original draft preparation, P.D. and Y.L.; writing—review and editing, P.D. and J.S.; visualization, S.X.; supervision, P.D.; project administration, P.D.; funding acquisition, J.S. and M.W. All authors have read and agreed to the published version of the manuscript.

**Funding:** This research was funded by the National Natural Science Foundation of China (32072203), the Innovation Fund of Haihe Laboratory of Synthetic Biology (22HHSWSS00013), Shanxi Provincial Department of Science and Technology (202204010931002, 2022ZDYF124), Key Research and Development Program of Shanxi, and Key Research and Development Program of Ningxia (2022BBF02010), Project of Tianjin Science and Technology Plan (21ZYJDJC00030).

**Institutional Review Board Statement:** Not applicable.

**Informed Consent Statement:** Not applicable.

**Data Availability Statement:** The data presented in this study are available in article and supplementary.

**Acknowledgments:** Thanks to Beijing Boxbio Science & Technology Co., Ltd. for providing chemical reagents and technical support.

**Conflicts of Interest:** The authors declare no conflict of interest.

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
