# Peer review of "Effect of Microbial Reinforcement on Polyphenols in the Acetic Acid Fermentation of Shanxi-Aged Vinegar"

_fermentation, doi:10.3390/fermentation9080756_

Round 1

Reviewer 1 Report

              The study titled “Effect of microbial reinforcement on polyphenols in the acetic acid fermentation of Shanxi aged vinegar” studied the relationships between the microorganisms and polyphenols. These results were interesting. However, some complementary information will support readers.

Major comments

              The journal “fermentation” is an international journal. Therefore, information about the “Shanxi aged vinegar” is required for non-Chinese readers. For example, the materials and processes of Shanxi aged vinegar production are required. In the Discussion section, the results for each fermentation stage are discussed. Thus, these stages must be defined.

              The resolution of the figures was insufficient for recognition.

Minor comments

All: The abbreviations must be defined when a word first appears. After definition, the word must be expressed as an abbreviation. Please unify the entire manuscript accordingly.

All: Scientific names are expressed in italics.

L89: When were samples collected?

L98: “ziplock” is trademark, isn’t it? Please, indicate it if necessary.

L122: Please indicate the manufacturer’s address.

L132–142: The programs and databases should be cited for their article. Please add the references.

Reviewer 2 Report

The name of the microorganism should be written in italics (abstract and other parts of the manuscript) - please correct that (remember about figures and tables).

Figures and charts are too small, which makes them difficult to read.

Citations are written carelessly in the text, contrary to the guidelines - please check the journal guidelines and correct that.

The introduction is poorly written, there are repetitions, it certainly should be expanded.

All the strains are previouslyscreened in the laboratory and stored in the Mi- 73 crobiology Laboratory of the School of Biological Engineering, Tianjin University of Sci- 74 ence and Technology. --> previously screened --> please read the whole text once again really carefully and correct the mistakes like this one.

The names of lactic acid bacteria are incorrect, for example: there is no anymore Lactobacillus plantarum, the right name is Lactiplantibacillus plantarum - please verify and correct that.

Reading the work, I get the impression that the text was written carelessly, there are a lot of language errors, typos, a lot of punctuation errors. All this makes the work illegible and difficult to read - especially result section.

The fact that the results are described and discussed meticulously and in detail certainly deserves praise - I have no objections.

Reviewer 3 Report

This manuscript describes that production of polyphenols can be increased by exogenous microbial addition in Shanxi aged vinegar. The strategy of experiment makes the study interesting, but the results do not support it well.

In this study, Spearman correlation analysis and single factor experiment were used to screen and verify the microbial factors. However, more important thing is that the results of microbial community analysis showing the dominance of these bacteria added during the acetic acid fermentation process should be presented to explain increase of polyphenols by changing microorganisms.

1.      The name of some Lactobacillus genus has been changed.

Ex) Lactobacillus plantarumLactiplantibacillus plantarum. Please change it throughout the entire manuscript.

2.      In Fig. 2, please explain that there is little change in pH despite a gradual increase of total acid.  

Round 2

Reviewer 3 Report

The characteristics of microorganisms appear differently depending on the strains. It has strain specificity. The strain selected should be accurately stated in the manuscript. In order to prove polyphenol increasing by the added strain, the results of microbial community analysis after adding this strain to vinegar fermentation should be presented.
